# Aminoacyl-tRNA Synthetases and tRNAs for an Expanded Genetic Code: What Makes them Orthogonal?

**DOI:** 10.3390/ijms20081929

**Published:** 2019-04-19

**Authors:** Sergey V. Melnikov, Dieter Söll

**Affiliations:** 1Department of Molecular Biophysics and Biochemistry, Yale University, New Haven, CT 06511, USA; sergey.melnikov@yale.edu; 2Department of Molecular Biophysics and Biochemistry, Department of Chemistry, Yale University, New Haven, CT 06511, USA

**Keywords:** synthetic biology, expanded genetic code, tRNA, aminoacyl-tRNA synthetases, orthogonal translation systems

## Abstract

In the past two decades, tRNA molecules and their corresponding aminoacyl-tRNA synthetases (aaRS) have been extensively used in synthetic biology to genetically encode post-translationally modified and unnatural amino acids. In this review, we briefly examine one fundamental requirement for the successful application of tRNA/aaRS pairs for expanding the genetic code. This requirement is known as “orthogonality”—the ability of a tRNA and its corresponding aaRS to interact exclusively with each other and avoid cross-reactions with additional types of tRNAs and aaRSs in a given organism.

In the past two decades, over a dozen natural and engineered tRNA molecules have been used to expand or rewrite the genetic code of a living cell. With the help of these tRNAs, numerous strains of bacteria and eukaryotes have been created that are capable of site-specific incorporation of more than 170 non-canonical amino acids into cellular proteins in vivo (reviewed in refs. [1,2,3]). Typically, these engineered organisms possess one or a few additional tRNAs and corresponding tRNA synthetases, which are known as “orthogonal” tRNA/tRNA synthetase pairs. These orthogonal pairs are designed or selected in such a way that they incorporate a corresponding non-canonical amino acid but do not cross-react with other amino acids, tRNAs and tRNA synthetases in a given organism. Therefore, engineering of new tRNA/tRNA synthetase pairs to expand the genetic code requires understanding of what makes these pairs orthogonal. 

Early studies revealed that specific tRNA recognition by a cognate tRNA synthetase relies on a small fraction of nucleotides in the tRNA structure (reviewed in refs. [4,5]). These nucleotides were called identity elements and anti-identity elements as they respectively trigger specific tRNA aminoacylation by a cognate tRNA synthetase and prevent tRNA mis-aminoacylation by non-cognate tRNA synthetases. Most typically, both identity and anti-identity elements are located at the two opposite sides of tRNA molecules: the anticodon stem and the acceptor stem. Typical examples of the identity elements include anticodon residues I34, A35, U36 in *Saccharomyces cerevisiae* tRNA^Ile^ or the acceptor stem residue A73 along with anticodon residues U34, U35, U36 in *Escherichia coli* tRNA^Lys^ [6,7,8]. The presence of base modification (e.g., thiolation leading to s^2^U34 in *E. coli* tRNA^Glu^) contributes significantly to tRNA identity, as lack of the modification decreases the aminoacylation efficiency drastically [9]. Typical examples of anti-identity elements include base modifications of *E. coli* tRNA^Ile^ residue C34 (*lysidine*), which prevents tRNA^Ile^ mis-aminoacylation by MetRS synthetase, and methylation of G73 base in *S. cerevisiae* tRNA^Asp^ (m^1^G), which prevents tRNA^Asp^ mis-aminoacylation by ArgRS synthetase [10,11]. Thus, although tRNA molecules typically comprise ~70–100 nucleotides, only a small fraction of these nucleotides is required for specific recognition of a tRNA by a corresponding tRNA synthetase. 

Discovery of the identity elements was empowering as it showed that the correspondence between tRNA molecules and tRNA synthetases can be reprogrammed by transplanting identity elements from one tRNA into another. For example, a transplantation of just three anticodon residues C34, U35, A36 from *E. coli* tRNA^Met^ to tRNA^Val^ switches the tRNA^Val^ specificity from ValRS to MetRS synthetases, leading to tRNA^Val^ acylation with methionine [12]. Similarly, transplantation of a single G3-U70 base pair from *E. coli* tRNA^Ala^ to tRNA^Phe^ [13,14] or tRNA^Cys^ [14] turn these tRNA^Phe^ and tRNA^Cys^ species into specific substrates of alanyl-tRNA synthetase, leading to their aminoacylation with alanine. Thus, discovery of the identity elements allowed to use relatively subtle changes in tRNA structures to redefine the rules of the genetic code.

Before synthetic biologists began to exploit tRNA identity elements to engineer and reprogram the genetic code, many experiments with tRNA identity elements were observed in nature. For example, the fungal pathogen *Candida albicans* was shown to have an unusual tRNA, tRNA^Ser^_CGA_, with hybrid identity elements that cause tRNA^Ser^_CGA_ recognition by two tRNA synthetases, LeuRS and SerRS [15]. Hence, the polysemous tRNA^Ser^_CGA_ can be aminoacylated either by leucine or serine, leading to ambiguous translation of CUG codons—a property that helps *C. albicans* to introduce statistical variation in protein sequences that regulate such virulence attributes as morphogenesis, phenotypic switching, and adhesion [16]. Also, tRNA identity elements were found in a number of tRNA-like molecules. For example, genomic RNA of the turnip yellow mosaic virus carries a stem-loop structure that mimics the tRNA^Val^ anticodon. This mimicry allows the viral RNA to recruit valyl-tRNA synthetase, aminoacylate the 3′-end of the viral RNA with valine, and eventually, initiate translation of viral RNA [17]. Similarly, in bacteria, tmRNA, a tRNA/mRNA hybrid that is used to rescue stalled ribosomes on truncated mRNAs [18], carries an identity element of tRNA^Ala^ (the base pair A3-U71) that allows it to aminoacylate tmRNA with alanine and use tmRNA to rescue stalled ribosomes [19]. Overall, these studies showed that specific recruitment of a tRNA synthetase can be achieved even in the absence of canonical tRNA structure, as long as tRNA-like molecules carry a proper set of identity elements. 

The first successful attempts to use tRNA identity elements to expand the genetic code of live cells exploited variability of identity elements in homologous tRNAs from different species. For example, in bacteria, tRNA^Tyr^ identity elements include the base pair G1-C72, whereas in archaea and eukaryotes tRNA^Tyr^ identity elements include the reversed base pair, C1-G72 (reviewed in ref. [20]). This difference allowed the transplant of the tRNA^Tyr^/TyrRS pair from the archaeon *Methanocaldococcus jannaschii* into *E. coli* (with a few modifications in the tRNA^Tyr^) that created a strain carrying an additional tRNA^Tyr^/TyrRS pair that does not cross-react with other tRNA synthetases/tRNAs in *E. coli*, and that allows the genetic encoding of an additional amino acid, *O*-methyl-l-tyrosine [21]. Similar variations between bacterial, archaeal, and eukaryotic tRNA homologs allowed the creation of over a dozen orthogonal tRNA/synthetase pairs for genetic code expansion (Figure 1), most typically by transplanting archaeal tRNA/synthetase pairs from archaea into bacteria and from bacteria into eukaryotes (reviewed in ref. [1]).

More sophisticated orthogonal tRNA/synthetase pairs were engineered by constructing chimeric tRNAs [22] or by focused tRNA mutagenesis and subsequent selection. Two typical examples of these strategies are tRNA^UTu^ [23] and tRNA^SecUX^ [24]—engineered tRNAs that were developed for genetic encoding of selenocysteine. tRNA^UTu^ was constructed as a hybrid between tRNA^Sec^ and tRNA^Ser^ carrying the determinants for EF-Tu binding [23]. Due to the presence of tRNA^Sec^ segments, Ser-tRNA^UTu^ is recognized by the protein SelA for conversion to Sec-tRNA^UTu^. However, due to tRNA^Ser^ segments, tRNA^Utu^ binds EF-Tu and is delivered to the ribosome—something that natural tRNA^Sec^ is not capable of doing. Another tRNA for the genetic encoding of selenocysteine, tRNA^SecUX^, was evolved from *E. coli* wild-type tRNA^Sec^_UCA_ and also selectively engineered with the EF-Tu determinants [24]. These examples illustrate that successful engineering of tRNA/synthetase pairs typically require manipulation of both identity and anti-identity elements in the tRNA structure. 

Finally, there are a few examples in which orthogonality of tRNA/synthetase pairs stems from a large number of idiosyncratic contacts at the tRNA/synthetase interface, rather than from a small number of identity and anti-identity elements in the tRNA structure. Perhaps the best example of this scenario can be found in the tRNA^Pyl^/pyrrolysyl-tRNA synthetase pair. Phylogenetic analyses indicate that a tRNA^Pyl^/PylRS pair could have originated from a tRNA^Phe^/PheRS pair about 3 billion years ago [25]. Structural studies show that tRNA^Pyl^ is recognized by PylRS in a highly unusual manner: unlike most other tRNA synthetases, PylRS does not bind the tRNA anticodon and instead makes extensive contacts with the acceptor stem and the unusually small variable loop of tRNA^Pyl^ [26,27]. In this case, the orthogonality appears to stem from the long-term coevolution of the tRNA and its corresponding tRNA synthetase. This long-term coevolution appears to create a remarkable complementarity of shape and charge between tRNA^Pyl^ and PylRS, which makes the tRNA^Pyl^/PylRS pair (e.g., from the archaeon *Methanosarcina mazei*) exceptionally orthogonal when transplanted into a wide range of bacterial or eukaryotic organisms. 

Where do we go from here? As described above, a range of orthogonal tRNA synthetase/tRNA pairs are now available and may be enhanced by additional laboratory evolution. What are the desired characteristics that drive further improvement? The current version of orthogonal tRNA synthetases has moderate catalytic activity and is polyspecific for many non-canonical amino acids [28]. Thus, increasing catalytic power and substrate specificity is of great interest; significant achievements have already been reported for *p*-azido-PheRS using multiplex automated genome engineering (MAGE) [29] and PylRS using phage-assisted continuous evolution (PACE) [30] or phage-assisted non-continuous evolution (PANCE) [27]. The finding of two classes of PylRS enzymes [31] has led to the development of mutually orthogonal PylRS/tRNA^Pyl^ variant pairs that are able to serve different codons with different non-canonical amino acids [32,33]. In addition, the recent finding that three mutually orthogonal tRNA synthetase/tRNA pairs allowed successful site-specific non-canonical amino acid insertion at three distinct sites in a recombinant protein [34] is a sign of the wide range of future applications of genetic code expansion. Also, a recently identified class of tRNA molecules, known as allo-tRNAs [35], and their successful use for selenoprotein synthesis [36], illustrates that there might be many more natural tRNAs and tRNA-type molecules that can serve as templates for the future development of robust and useful orthogonal tRNA synthetase/tRNA pairs for the expanded genetic code. 

## Figures and Tables

**Figure 1 ijms-20-01929-f001:**
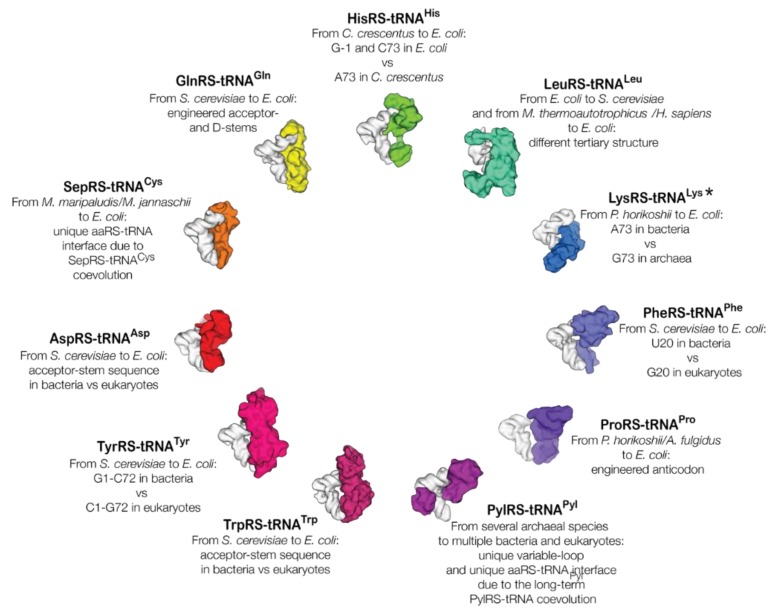
tRNA molecules and corresponding aminoacyl-tRNAs that have been used to expand the genetic code of live cells. Crystal structures of tRNA molecules (in white) are shown in complex with the corresponding aminoacyl-tRNAs (in the colors of the rainbow). The labels next to each structure indicate the origin of the orthogonal tRNA/tRNA synthetase pair and an organism into which this pair has been transplanted, which is followed by a brief description of tRNA identity or anti-identity elements that make a tRNA/tRNA synthetase pair orthogonal in an organism into which this pair has been transplanted. For example, the label “TyrRS-tRNA^Tyr^—From *S. cerevisiae* to *E. coli*: G1-C72 in bacteria vs C1-G72 in eukaryotes” means that the TyrRS-tRNA^Tyr^ pair has been transplanted from *S. cerevisiae* into *E. coli* where this pair remains orthogonal due to different identity elements between bacterial and eukaryotic tRNA^Tyr^: the G1-C72 pair in bacterial tRNA^Tyr^ and the C1-G72 pair in eukaryotic tRNA^Tyr^. The asterisk next to the LysRS-tRNA^Lys^ complex indicates that this model was produced by docking the crystal structure of tRNA^Lys^ into the tRNA-binding pocket of LysRS. The most comprehensive list of orthogonal tRNA/tRNA synthetase pairs and their classes can be found in the Supplemental files of Ref. [1].

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
