# Peer review of "Aminoacyl-tRNA Synthetases and tRNAs for an Expanded Genetic Code: What Makes them Orthogonal?"

_ijms, 2019, doi:10.3390/ijms20081929_

Round 1

Reviewer 1 Report

This short review is a nice compilation of identity/anti-identity elements for aminoacyl-tRNA synthetase (aaRS)-tRNA pairs and how engineered and naturally-occuring orthogonal pairs have been used in expanding the genetic code. There is much more that could be said, of course (for example more detail on selection of orthogonal tRNAs using randomized transcripts and negative selection with suppression of stop codons in barnase then positive selection with stop codons in beta galactosidase) - but this is a good brief addition to the field.

There is some cleanup of language required. For example

Line 23 "non-canonical amino acid into" should be "acids".

Line 43 "Thus, despite..." should probably be "Thus, although"

Line 54 "the identify elements allowed researchers to..." seems to be missing a word

Line 63 no comma after atributes

Line 69 "allows it to aminoacylate" missing "it"

Line 78 "allowed transplant of" instead of allowed to transplant

Line 82 "create over a dozen orthogonal" ("of" not needed)

Line 96 "typically require manipulation" instead of second engineering

Line 114 "that have been used to expand"

Line 139 comma before "illustrates" instead of after

Author Response

Thank you very much for your critical feedback. We have incorporated all your suggestions in the text and hope that we could improve our English. Thanks again for your time and correction!

Reviewer 2 Report

This concise and well-written mini-review by Melnikov and Söll discusses the fundamental requirement of orthogonality for the successful application of tRNA/aaRS pairs for expanding the genetic code. Natural tRNA/aaRS pairs can already be seen as “orthogonal” to one another in order to prevent mis-aminoacylation. By taking advantage of some of the underlying structural principles of natural tRNA/aaRS pairs, it has been possible to derive new orthogonal pairs but other strategies described in the review have also been developed.

It would be most helpful if the authors would indicate the class (class I or class II) to which each of the aaRS displayed in Figure 1 belongs. The authors should also discuss how the usage of aaRS belonging to different aaRS classes could be beneficial for creating new orthogonal tRNA/aaRS systems.

In addition to Figure 1, it would be useful to provide a table with an exhaustive list of the different rules, design principles and strategies used to create orthogonal tRNA/aaRS pairs. One of the table columns can also list the corresponding cited references.

Overall, this mini-review is very interesting and should be positively considered for publication.

Author Response

Dear reviewer, 

first of all, thank you very much for your time, and critical feedback!

To address your comments, we have modified figure 1 and added the information about the class I/class II of the synthetases. We then provided a reference to the review (Mukai et al.) that provides the table with the most comprehensive list of orthogonal tRNA/aaRS systems and how they were designed. We hope you will find these changes sufficient for this manuscript to be accepted for publication.

Yours, 

the manuscript's authors